# Trends and Determinants of EBF among Adolescent Children Born to Adolescent Mothers in Rural Bangladesh

**DOI:** 10.3390/ijerph17249315

**Published:** 2020-12-12

**Authors:** Aminur Rahman, Dilruba Nomani, Surasak Taneepanichskul

**Affiliations:** 1College of Public Health Sciences, Chulalongkorn University, Bangkok 10330, Thailand; surasakta@yahoo.com; 2International Centre for Diarrheal Disease Research (ICDDR,B), Dhaka 1212, Bangladesh; nomani.dilruba@icddrb.org

**Keywords:** adolescent, exclusive breastfeeding, Bangladesh

## Abstract

Exclusive breastfeeding (EBF) has proven benefits for both mothers and infants; however, adolescent mothers have poor EBF practices globally. In Bangladesh, the practice of EBF among adolescent mothers remains unexplored. The aim of our study was to understand the EBF practices among adolescent mothers and their determinants in both the Health and Demographic Surveillance (HDSS) system areas of the International Centre for Diarrheal Disease Research, Bangladesh (icddr,b) service area (ISA) and government service area (GSA) in rural Matlab, Bangladesh. For the purpose of our study, we collected relevant information from the database of the Health and Demographic Surveillance System (HDSS) of icddr,b and performed analysis to understand the trends and identify the determinants of EBF and identify the determinants of EBF among adolescent mothers living in two areas between 2007 and 2015. In total, 2947 children born to adolescent mothers were included in our final analysis. We used the Kaplan–Meier and the Cox-proportional hazards models to determine the differences in EBF practices in the two areas. We noted a lower trends of EBF in the ISA compared to the GSA in bivariate analysis. However, after adjusting for confounding variables, EBF status was 15% lower in the GSA than the ISA (HR: 0.85, 95% CI: 0.72–0.99). The father’s education was significantly different among the two populations. In both study areas, the coverage of EBF among adolescent mothers was lower than the national average (42% vs. 65%). Ensuring that adolescent mothers receive optimal care may improve EBF behavior.

## 1. Introduction

Breastfeeding is known to be the preeminent nutrition to save lives, prevent illness, and ensure healthy development for the first 1000 days of life [1]. Exclusive breastfeeding (EBF) has well documented benefits for both mothers and infants. The breastfeeding practices among adolescent mothers is the lowest globally [2] despite an increasing proportion of adolescent mothers [3]. Adolescent pregnancy (between the ages of 10 and 19 years of age) [4,5] contributes to 10% of all annual maternal deaths and higher levels of infant mortality and morbidity [6,7]. In low and middle income countries, breastfeeding is estimated to prevent around 13% of all deaths among children under five years of age. Several beneficial effects of EBF have been documented worldwide, such as reduced postnatal mortality rates and sudden infant death syndrome, lower risk of childhood infection, and reduced chance of diabetes [8]. Maternal breastfeeding benefits include lower risk of developing breast and ovarian cancers, adequate weight recovery, and lactational amenorrhea, which could be a natural birth control [9]. Consequent to these proven benefits, breastfeeding was supported by “The Global Strategy on Infant Young Child Feeding” approved by the 55th World Health Assembly (WHA) in 2003 [10]. The World Health Organization (WHO) recommends the EBF of infants for the first six months of infant lives. WHO defines EBF as receiving only breast milk without any additional food or drink [9]. However, its practice is limited and, in fact, its reduction is considered a serious problem, especially in developing countries.

EBF practices among adolescent mothers varies around the world. The rates of EBF among infants younger than six months of age were reported at 20% in the central and eastern European countries and 44% in south Asia [11,12]. However, only half of the neonates (first 30 days of life) and 30% of the infants aged 1–5 months were exclusively breastfed globally [13]. In Thailand, the practice of breastfeeding by adolescent mothers is considered to be poor [14]. In Ecuador it is higher than the national average (43% vs. 63%) [15]. A higher proportion of young (50.2%) and older mothers (63.5%) in Ontario, Canada, breastfed their children [16]. The breastfeeding practice is a complex issue, since it is influenced by personal, biological, socio-cultural, and environmental factors [17,18]. In the United Kingdom (UK) younger mothers <20 years age, are less likely to breast feed their children than older mothers, since they lack parenting experience, are less educated, and most are from disadvantaged groups [18]. However, the factors that are associated with low rates of breastfeeding among adolescents are not well understood. A study identified adolescent pregnancy as a factor in the discontinuation of EBF [19]. The duration of breastfeeding among adolescent mothers may be associated with demographic characteristics such as educational level and multiple births [20,21]. A study noted previous pregnancies to be associated with higher likelihood of choosing breastfeeding by adolescent mothers [21]. 

In Bangladesh, adolescent mothers believed that EBF for infants up to six months of age means that they may receive water and other liquids along with breast milk [22]. In Bangladesh, breastfeeding is common but the rate of EBF is very low [23]. According to the Bangladesh Demography and Health Survey (BDHS) 2016 [24], 65% of infants younger than 6 months were exclusively breastfed, but EBF practices among adolescent mothers was not reported [25] despite potentially higher vulnerability of younger mothers and their children associated with poverty and higher rates of early marriage and childbearing. In Bangladesh, 65% of children less than 6 months of age are exclusively breastfed according to Bangladesh Demographic and Health Survey (BDHS) 2016. However, no such information is available about adolescent mothers’ EBF practices despite high potential vulnerability of younger mothers and their children due to poverty and high rates of early marriage and childbearing. This is important and would be the first instance in Bangladesh where a retrospective longitudinal study of adolescent mothers and their children conformed to the recommended breastfeeding advice in the HDSS Matlab area (both in the International Centre for Diarrheal Disease Research, Bangladesh (icddr,b) service areas (ISA) and the government service area (GSA)) of Bangladesh. The aims of this study were to find the prevalence of exclusive breastfeeding among adolescent mothers to identify the factors influencing EBF practices. These findings will help to assess the trends of EBF in both study areas, understand the factors influencing EBF practices in rural Bangladesh to some extent, and of course identify the challenges to improving EBF practices among adolescents. Further, these findings could contribute to developing specific program strategies and policy making for this specific vulnerable age group of mothers of rural Bangladesh and other similar contexts; without ensuring their health security, universal health coverage cannot be achieved.

This is the first report on EBF practices among adolescent mothers living in the Health and Demographic Surveillance System (HDSS) areas, comparing them with those living in the government service areas (GSA) in Matlab, rural Bangladesh. The aims of this study were to find the prevalence of EBF among adolescent mothers and to identify the factors influencing EBF practices.

## 2. Materials and Methods

### 2.1. Study Design

We followed a retrospective longitudinal study design. In this study, we used data from the Health and Demographic Surveillance System (HDSS) of the International Centre for Diarrheal Disease Research, Bangladesh (icddr,b) to address the study objectives. The study objective was to compare the prevalence of EBF among adolescent mothers in icddr,b and government service areas in Matlab and to investigate their socio-demographic determinants.

### 2.2. Study Population

Children of adolescent mothers who were breastfeeding in the HDSS databases during the 2007 to 2015 period constituted the study population for our research. Children of adolescent mothers with missing breastfeeding information during the study period (2007–2015) were excluded from the study. In total, 2947 children of adolescent mothers who were breastfed were included in the study; of them, 1711 were from the icddr,b service area (ISA) and 1236 from the government service area (GSA).

### 2.3. Study Setting

Since 1966, the Health and Demographic Surveillance System (HDSS) has been operating in a rural area of Bangladesh named Matlab. As per Figure 1, the Matlab HDSS area was divided into two parts in 1987: the icddr,b service area (ISA: administrative blocks A, B, C, and D), and the government service area (GSA: administrative blocks E, F, and G), covering 142 villages. HDSS has been pulling together vital statistics (live births, still births, miscarriages, deaths, marriages, and in and out migration) through community health research workers (CHRWs) since 1966 [25]. The CHRWs accumulate vital demographic data by visiting each household on a bi-monthly basis. At each visit CHRWs complete vital event registration forms. The details are described elsewhere [26,27].

Data collection methods are described elsewhere in these studies [26,27,28].

### 2.4. Quality of the Data

At the beginning of the year the service area for each CHRW is assigned by icddr,b. Every month, on a certain day, all CHRWs (both the icddr,b and government service areas) sit together to update their own registrar books. The assigned supervisors make routine spot checks of 2% of the samples. All cleaned data are stored within the longitudinal data system and checked with a set of validation criteria before final storage after going through three tiers of supervision by Field Research Supervisors (FRSs), Field Research Officers (FRO), and a senior manager and are then processed through an error detecting computer program.

### 2.5. Data Analysis

Data was analyzed using SPSS 23 statistical software. The outcome variables were the practice of exclusive breastfeeding among adolescent mothers. EBF is described as feeding where a newborn does not take anything other than the mother’s breast milk until 6 months of age, except saline, which is considered a medicine, and medicine when prescribed. The independent variables covered socio-demographic factors. Economic status was measured in terms of relative asset quintiles rather than in terms of income or consumption [29,30]. Asset quintiles were constructed using asset variables from the socio-economic survey of year 2014 using principal component analysis (PCA) and factor analysis methods. Socio-demographic differences between the two service areas were measured by chi-square test. The distribution of the practice of EBF among adolescent mothers from 2007 to 2015 was explored in both areas. The predictors associated with practicing EBF were identified with the Cox proportional hazards model and adjusted for socio-demographic variables. Statistical significance was defined as *p*-values < 0.05.

## 3. Results

There were 2947 cases of adolescents mothers who breastfed their children in the HDSS area during 2007 to 2015. Using live births as the denominator in these two areas, on average, the percentage of EBF in the ISA was 42%, whereas for the GSA it was 44%. The percentage of adolescent mothers practicing EBF was 43 in the ISA and 46 in the GSA in 2007. Percentages of EBF in both service areas were reduced at the end of the study period (2015) from the initial time point (2007), although the decrease over time was not statistically significant (*p* < 0.05) (Figure 2). Figure 2 shows that throughout the study period, the percentage of adolescent mothers practicing EBF was lower for icddr,b compared to the government area, except for the year 2010. The difference was slightly greater in 2013 compared to other years, when the percentage of adolescent mothers practicing EBF was 42 in the icddr,b area and 54 in the government service area. In 2015, the percentages in the icddr,b and government service areas almost converged (Figure 2).

The socio-demographic characteristics of adolescent mothers practicing EBF are described in Table 1.

Table 1 shows that amongst 2947 adolescent mothers practicing EBF in both the icddr,b service area (ISA) and government service area (GSA), the distribution of categories of mother’s age at first birth across ISA and GSA was found to be nearly even. More than 95% had completed at least primary education or higher in both ISA and GSA. Around 55% of husbands of adolescent mothers had completed the same. The percentage of boys was found to be higher in both areas compared to girls. In both areas, adolescent mothers were predominantly Muslim, and most of the adolescent mothers had a parity of 1. More than 70% of deliveries were normal in both areas. Facility deliveries were double in the icddr,b service area (ISA) relative to the government service area (GSA). Approximately 14% adolescent mothers had taken 4+ ANC (mother who had received 4 or more antenatal care visits) in both areas (Table 1).

A significant difference between ISA and GSA (log rank test < 0.0222) was found in terms of the time of EBF. The survival probabilities for EBF among adolescent mothers from the GSA were significantly higher compared to those of the ISA, as per Figure 3.

The determinants associated with the event of discontinuing EBF in the ISA and GSA and the findings from multivariate analysis using the Cox proportional hazards model are exhibited in Table 2.

Bivariate findings in Table 2 demonstrated that there were a total of 824 cases of adolescent mothers who discontinued EBF from both the ISA and GSA. Only the government service area and paternal education were found to have a significant relation with the practice of EBF among adolescent mothers (*p* < 0.05); 29.6% of adolescent mothers in the ISA discontinued EBF of their children, whereas in the GSA, 25.6% of adolescent mothers did the same. Less than 30% of children also had experience with the discontinuation of EBF. The distribution of categories of other covariates was found to be non-significant with the discontinuation of EBF as per bivariate findings (Table 2). The event of discontinuing EBF among adolescent mothers was measured by adjusting the effect of area (government service area or icddr,b area), maternal education, paternal education, asset score, religion, delivery place, mode of delivery, number of antenatal care (ANC) visits, adolescent mothers’ age at birth, sex of children, and repeated pregnancy during the study period. These covariates were selected based on their significant relationship with discontinuation of EBF (from bivariate analysis findings and literature review). The incidence of discontinuing EBF was lower for an adolescent mother residing in the government service area (AHR: 0.85, 95% CI: 0.72–0.99) compared to an adolescent mother residing in the icddr,b area, keeping all the other variables at a fixed level (Table 2). This result was different from the findings from Kaplan–Meier survival estimates (Figure 3), where adolescent mothers of the GSA had significantly higher possibilities of discontinuing EBF compared to those of the ISA. Adolescent mothers with no education (AHR = 1.02, 95% CI: 0.63–1.66) or primary education (AHR = 1.03, 95% CI: 0.85–1.24) were found to be more likely to stop EBF compared to highly educated adolescents for this study (this findings was not significant). Cox proportional hazards regression analysis also revealed no gender differences to being exclusively breastfed (AHR: 1.00, 95% CI: 0.88, 1.15). Adolescent mothers from families of higher socio-economic status (AHR: 0.84; 95% CI: 0.67, 1.07) had a lower risk of early cessation of EBF compared to adolescent mothers from lower socio-economic status. Similarly, adolescent mothers who had 4+ ANC visits (AHR = 1.03, 95% CI: 0.84–1.25) were more prone to stopping EBF relative to who had less than 4 ANC visits, but this findings was not significant. Adolescents who had delivered their babies at home (AHR = 0.98, 95% CI: 0.82–1.17) were less likely to stop EBF compared to those who went to health facilities for delivery of their baby. Findings also stated that adolescent mothers who delivered their babies through C-section were less likely to stop EBF compared to those who had a normal delivery.

## 4. Discussion

### 4.1. Study Summary

This study shows that the prevalence of EBF was 42% for the ISA and 44% for the GSA for the study period from 2007 to 2015, which are similar to findings from other regions such as the Turk Islands (50.6%) [31] and Sub-Saharan African countries (41.0%) [32], but far below the national average [33]. Bivariate findings revealed that father’s education, religion, parity, and place of delivery were significantly different among the two populations (*p* < 0.05). Although many factors are related to EBF among adolescent mothers worldwide, the findings of this study indicated that the key factor contributing to breastfeeding practice is service area. In spite of having the Maternal, Newborn, and Child Health–Family Planning (MNCH–FP) program [27] in the icddr,b area, the practice of EBF was found to be higher in the government service area after adjusting for other covariates, such as child sex, mother’s age at first birth, maternal education, paternal education, religion, asset score, place of delivery, mode of delivery, history of 4+ ANC visits, and repeated pregnancy, which were incorporated based on their significant association with EBF practices and the literature review.

### 4.2. Socio-Demographic Discussion

Bivariate analysis shows that excusive breastfeeding status in the ISA is better than in the GSA, but after adjustment for confounding variables (child sex, mother’s age at first birth, maternal education, paternal education, religion, asset score, place of delivery, mode of delivery, history of 4+ ANC visits, repeated pregnancy) the findings shift in the opposite direction. This indicates the requirement of much stronger service delivery and knowledge sharing in terms of infant feeding in both the ISA and GSA. The current study revealed different scenario from study findings across different countries of the world [34]. One study showed that mothers who had higher education were less likely to practice EBF than those with no education [35], which is different from our study findings, although our result was not significant. In this study, it was observed that fathers were less educated than mothers, although another study of the Matlab area of Bangladesh showed that percentage of no education was higher for women (19%) than men (14%) [36]; however, as our study was based on a retrospective cohort of adolescent mothers only who breastfed, consequently only data for their husbands’ education were used, which could be one of the reasons that many more fathers had no education compared to the adolescent mothers. In addition, this could be due to the fact that men usually get married early when they are not educated enough, and a good number of them left for the Middle-East for better income and were not able to continue their education; however, women continued their education even after the departure of their husbands. Adolescent mothers from higher socio-economic classes were less likely to stop EBF compared to adolescent mothers from lower economic classes of Matlab, which is similar to another study [37,38], but this result contradicts the findings of other earlier studies [33,39,40]. This might be due to lack of nutrition security among adolescent mothers from lower economic classes. Children who were delivered through C-section were exclusively breastfed compared to children born from normal delivery, which is similar with another context [39], but different from an earlier study conducted in Bangladesh in 2014 [38,40]. A few studies revealed that mothers who had taken antenatal care were more prone to practice EBF [24,41]. However, our study findings showed just the opposite scenario, similar to another study [40], which could be due to the fact that running ANC programs that include breastfeeding counseling are not robust or adequate enough to improve breastfeeding knowledge of mothers and motivates them to exclusively breastfeed their infant. A study in Nigeria revealed that female infants were more likely to be exclusively breastfed than their male counterparts [42], whereas a few studies showed just the opposite [43], which was not the case for the current study. Both male and female children were found to breastfed equally in this particular study, although the result was not statistically significant. A study finding showed that infants born in health facilities were about four times more likely to be exclusively breastfed than those who were born at home [44], which is different than our case. All the studies cited were concentrated on EBF behavior of mothers of different ages from different countries, and few were similar countries to Bangladesh or India, but none of these were focused on adolescent mothers, which could be a reason for the contradictory results. The reason behind this kind of variation in results from other studies might be caused also by other factors such as differences in study settings, the sample size of other studies being smaller than in our study, as well as factors that were not included in the study, like infant age, mother’s employment status, knowledge about good breastfeeding practices, monthly household income, mother’s smoking status, positive attitudes towards EBF, intent to exclusively breastfeed before delivery, infant birth weight, health system practices, discarding colostrum, community beliefs, etc. [38]

### 4.3. Way Forward

A further understanding of the determinants of EBF practice among adolescent mothers is necessary in these areas to develop policies that will improve breastfeeding rates and allow the World Health Organization’s goal of 90% EBF at six months to be reached. To achieve that, parents need to be educated about the benefits of colostrum and EBF, and the harmful effects of pre-lacteal feeding (giving fluid or semisolid food before breastfeeding to an infant during the first 3 days after birth or a mother who gives any food/fluid without the breast milk regardless of the frequency is considered as pre-lacteal feeding) [23]. In addition, to identify relevant factors influencing EBF, practical education-based interventions are needed. Strategies should be developed to assist adolescent parents to address the identified challenges to practices of EBF, rather than just giving out information on EBF. Ensuring that adolescent parents receive optimal maternity care can improve breastfeeding exclusivity rates, which ultimately will lead to improved maternal and child health outcomes. Strengthening infant feeding counselling during ANC follow-up and birth, promoting institutional delivery, and enabling every mother to encourage colostrum feeding were recommended in order to increase the proportion of women practicing EBF [44].

## 5. Strengths and Limitations

Both male and female children were found to breastfeed equally in this particular study, although the result was insignificant. A study finding showed that infants born in health facilities were about four times more likely to be exclusively breastfed than those born at home [44], which is different than our case. The reason behind such insignificant results from our study might be due to smaller sample size. To verify the impact of all potential variables, a larger sample size may be required. Data from the Matlab HDSS has been condemned for not being representative of other rural areas of Bangladesh because of its many and long-term interventions in the field of health, population, and nutrition [45]. It has been noted that that GSA CHRWs have a much larger catchment population than ISA CHRWs, which may result in less robust GSA data. The quality and robustness of this surveillance data is the main strength of this paper. The rigor of the data quality procedures and long standing follow up of the HDSS has provided a unique opportunity to produce authentic results (nearly no chances of information bias) from the analysis [26].

## 6. Conclusions

Education, wealth index, and residing in the ISA seem to be important determinants. These important factors need to be considered to improve EBF in the GSA. Policies, programs, and research should focus on improving breastfeeding strategies to support breastfeeding of infants. Without addressing the challenges regarding EBF practice among adolescents, the target of achieving universal health coverage by 2030 [46] cannot be accomplished.

## 7. Implications and Contributions

This is the first time that measurement of EBF of children among adolescent women in Bangladesh has been conducted. The prevalence of EBF among adolescent mothers’ children is very low, especially in the GSA. Policymakers should understand the reasons why in the ISA, EBF is higher than in the GSA and what needs to be taken into consideration to improve the EBF status of adolescent mothers’ children.

## 8. List of Abbreviations

ANC: antenatal care, BDHS: Bangladesh Demographic And Health Survey, CHRW: Community Health Research Worker, CI: confidence interval, EBF: exclusive breastfeeding, FRO: field research officer, FRS: field research supervisor, FVR: family visit record, FWC: family welfare center, FWV: family welfare visitor, GSA: government service area, HDSS: Health and Demographic Surveillance System, HR: hazard ratio, icddr,b: International Centre for Diarrheal Research, Bangladesh, ISA: icddr,b service area, MNCH–FP: Maternal, Newborn, and Child Health and Family Planning, MNCH: Maternal Neonatal and Child Health, OGSB: Obstetrics and Gynecology Society of Bangladesh, PNC: postnatal care, SDG: sustainable development goal, SPSS: Software Package for Social Statistics, SRB: service record book, TT: tetanus toxoid, UHC: Upazila Health Complex, UK: United Kingdom, WHA: World Health Assembly, WHO: World Health Organization.

## 9. Ethics Approval and Consent to Participate

The institutional review committee (IRC) at icddr,b provided ethical clearance for this analysis (protocol number: PR#17087). Data was accessed in compliance with icddr,b published data policies. The confidentiality and anonymity of study participants were strictly maintained. Data were presented in such a way so that any individual person could not be identified or traced back through the reported presentation of the information. The IRC covered both ISA and GSA in their observations.

## 10. Availability of Data and Materials

Data contain potentially identifying or sensitive information from delivering women. However, “Data can be available on request”. The data request should be submitted to the Research Administration (RA) of icddr,b and will be assessed by the corresponding ethics committee, namely the Institutional Review Board of icddr,b. As supplementary information, we have added approved protocol where one can obtain the study title and protocol number (PR-17087) against which a data access application should be made. Please visit https://www.icddrb.org/dmdocuments/icddrb%20Data%20Access%20Policy.pdf for additional information. Data requests are evaluated by icddr,b’s Data Repository Committee (DRC), and the Research Administration (RA) serves as the Secretariat of the DRC. The key contact person of RA at present is Ms. Armana Ahmed, Lead (A), RA at aahmed@icddrb.org. If the data request is considered justifiable by the DRC, then the RA will share the anonymous data with the applicant. Moreover, for any particular clarification of the research findings documented in this article, queries can be directed to the primary author of this article or to the corresponding author. Both of them can be contacted at draminur@icddrb.org. The email correspondence regarding data access can be addressed to the executive director office at dircetor@icddrb.org.

## Figures and Tables

**Figure 1 ijerph-17-09315-f001:**
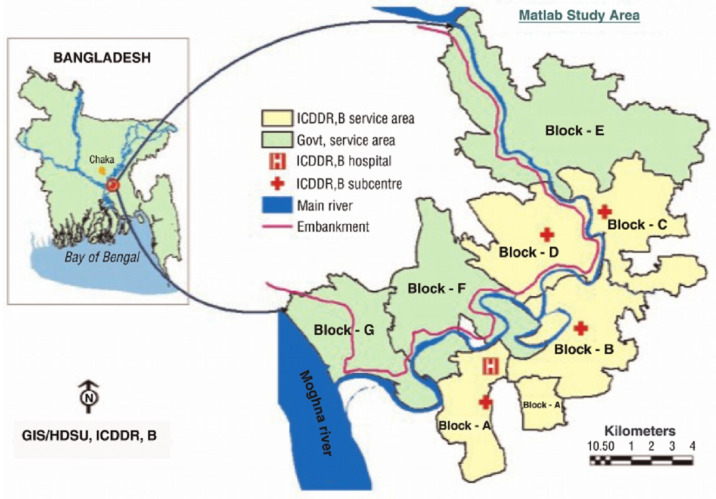
Map of the study area.

**Figure 2 ijerph-17-09315-f002:**
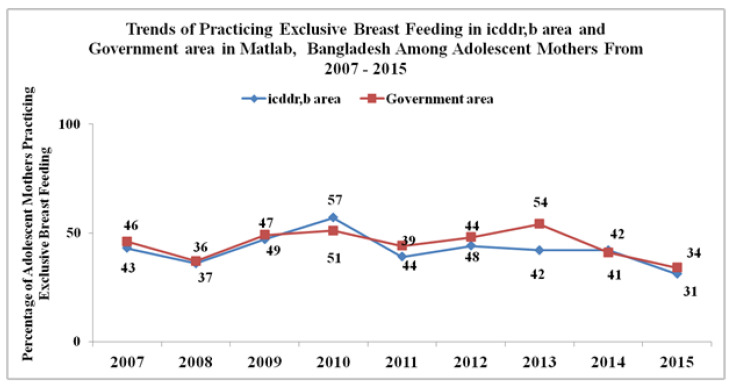
Trends of practicing exclusive breastfeeding (EBF) in the International Centre for Diarrheal Disease Research, Bangladesh (icddr,b) area and the government service area in rural Matlab, Bangladesh, among adolescent mothers from 2007 to 2015.

**Figure 3 ijerph-17-09315-f003:**
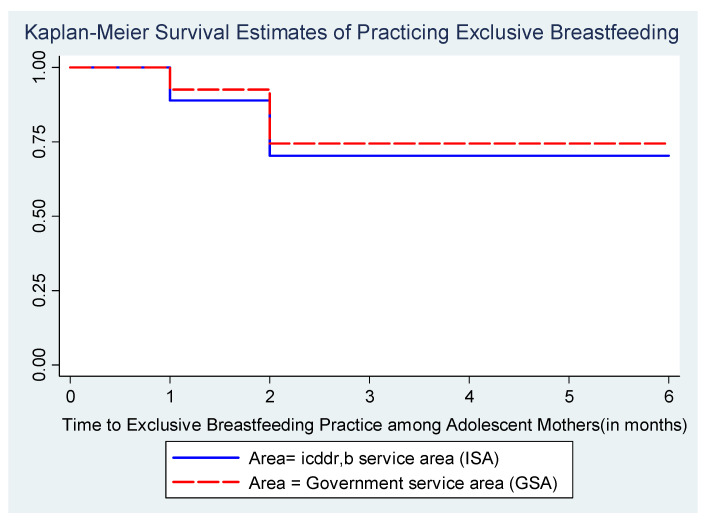
Kaplan–Meier survival estimates for time to EBF in icddr,b service area (ISA) and government service area (GSA).

**Table 1 ijerph-17-09315-t001:** Socio-demographic characteristics of adolescent mothers practicing EBF in both icddr,b service area (ISA) and government service area (GSA).

Socio-Demographic Variables	ISA (1711)	GSA (1236)	*p*-Value
	n (%)	n (%)	
Mother’s age at first birth			
≤17	379 (22.2%)	266 (21.5%)	0.259
18	488 (28.5%)	387 (31.3%)	
19	844 (49.3%)	583 (47.2%)	
Child sex			
Male	870 (50.8%)	624 (50.5%)	0.846
Female	841 (49.2%)	612 (49.5%)	
Maternal education			
No education	31 (1.8%)	28 (2.3%)	0.087
Primary	262 (15.3%)	223 (18.0%)	
Above primary	1418 (82.9%)	985 (79.7%)	
Paternal education			
No education	770 (45.0%)	541 (43.8%)	0.024 *
Primary	331 (19.3%)	289 (23.4%)	
Above primary	610 (35.7%)	406 (32.8%)	
Religion			
Islam	1533 (89.6%)	1171 (94.7%)	<0.001 *
Hindu	178 (10.4%)	65 (5.3%)	
Asset Score			
Lowest	258 (15.1%)	182 (14.7%)	0.897
Second	318 (18.6%)	227 (18.4%)	
Middle	302 (17.7%)	233 (18.9%)	
Fourth	414 (24.2%)	305 (24.7%)	
Richest	419 (24.5%)	289 (23.4%)	
Number of Parity			
1	1650 (96.4%)	1169 (94.6%)	0.015 *
2+	61 (3.6%)	67 (5.4%)	
Place of delivery			
Home	329 (19.2%)	730 (59.1%)	<0.001 *
Facility delivery	1382 (80.8%)	506 (40.9%)	
Mode of delivery			
CS	474 (27.7%)	313 (25.3%)	0.15
Normal	1237 (72.3%)	923 (74.7%)	
History of 4+ ANC			
Less than 4 visits	1465 (85.6%)	1070 (86.6%)	0.464
Four or more visits	246 (14.4%)	166 (13.4%)	

* indicates that the results are significant at *p* < 0.05.

**Table 2 ijerph-17-09315-t002:** Factors associated with exclusive breastfeeding: results from bivariate and multivariate analysis.

	EBF Discontinuation		Adjusted Hazard Ratio (AHR)	95% CI
	Event (N = 824) (n%)	*p*-Value		
Area				
GSA	317 (2 5.6%)	<0.001 *	0.85	0.72–0.99
ISA	507 (29.6%)		Ref	
Child sex				
Male	417 (27.9%)	0.952	1.00	0.88–1.15
Female	407 (28.0%)		Ref	
Mothers age at first birth				
≤17	169 (26.2%)	0.492	0.93	0.78–1.12
18	253 (28.9%)		1.05	0.89–1.23
19	402 (28.2%)		Ref	
Maternal education				
No education	17 (28.8%)	0.764	1.02	0.63–1.66
Primary	142 (29.3%)		1.03	0.85–1.24
Secondary and higher	665 (27.7%)		Ref	
Paternal education				
No education	336 (25.6%)	<0.001*	0.89	0.76–1.05
Primary	200 (32.3%)		1.11	0.91–1.34
Secondary and higher	288 (28.3%)		Ref	
Religion				
Islam	744 (27.5%)	0.072	Ref	
Hindu	80 (32.9%)		1.15	0.91–1.45
Asset Score				
Lowest	136 (30.9%)	0.088	Ref	
Second	168 (30.8%)		1.02	0.81–1.28
Middle	144 (26.9%)		0.89	0.70–1.13
Fourth	201 (28.0%)		0.94	0.75–1.17
Richest	175 (24.7%)		0.84	0.67–1.07
Place of delivery				
Home	288 (27.2%)	0.488	0.98	0.82–1.17
Facility delivery	536 (28.4%)		Ref	
Mode of delivery				
CS	214 (27.2%)	0.575	0.94	0.79–1.12
Normal	610 (28.2%)		Ref	
History of 4+ ANC				
No	705 (27.8%)	0.653	Ref	
Yes	119 (28.9%)		1.03	0.84–1.25
Repeated pregnancy				
Multiple	11 (27.5%)	0.948	1.03	0.56–1.89
Single	813 (28.0%)		Ref	

Note: * indicates that the results are significant at *p*-value < 0.05.

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
