# Peer review of "Trends and Determinants of EBF among Adolescent Children Born to Adolescent Mothers in Rural Bangladesh"

_ijerph, 2020, doi:10.3390/ijerph17249315_

Round 1
Reviewer 1 Report
The paper provides high quality analyses on the coverage of EBF practices among adolescent mothers and their determinants. I have included comments and suggestions below:
Abstract is not sufficient, for example, the result of father education, which was statistically significant, but not presented.
Line 75-77: the study aims need to be more specific, is different with study design, line 82-84,
Line 115: Exclusive breastfeeding: the reason of the except saline,
Figure 2, It's not enough to just compare the numbers, statistical value is required.
Line 150-151: female children is girls? Male children is boys?
Line 154: ANC, Number of Antenatal Care, abbreviation needs to be explained first.
Line 160: exclusive breastfeeding practice, EBF, the whole manuscript needs to be revised.
Line 177: male and female children? All children?
Less than 30% of male and female children experienced the same “what”?
Line 178: what is that meaning of “almost even”
Line 172-199 is the same as table 2, this part needs to be refined.
Line 228-235: the results found fathers and mothers needed the same education about exclusive breastfeeding, therefore this discussion part deviated from the research results.
Discussion: it is better to divide into three paragraphs, 1st paragraph is the summary of the study, 2st and 3st paragraph discuss geography and parental education, respectively
Line 224-225 could be moved into limitation.
6. Conclusions 7. Implication and Contribution are not sufficient, and should focus on the results of this current study.
Line 247: double that, editing and language revisions of the whole manuscript are necessary.
Author Response
Reviewers 1:
The paper provides high quality analyses on the coverage of EBF practices among adolescent mothers and their determinants. I have included comments and suggestions below:
Abstract is not sufficient, for example, the result of father education, which was statistically significant, but not presented.
Response: done with line 16-17
Line 75-77: the study aims need to be more specific, is different with study design, line 82-84,
Response: Study design corrected at line 75
Line 115: Exclusive breastfeeding: the reason of the except saline,
Response: Corrected at line 117-118
Figure 2, It's not enough to just compare the numbers, statistical value is required.
Response: Corrected in line with 176-178
Line 150-151: female children is girls? Male children is boys?
Response: Done at line 152-153
Line 154: ANC, Number of Antenatal Care, abbreviation needs to be explained first.
Response: Done at line 157
Line 160: exclusive breastfeeding practice, EBF, the whole manuscript needs to be revised.
Response: exclusive breast feeding converted as EBF in whole manuscript; line: 163,164,176,179,181,183,187,209, 315
Line 177: male and female children? All children?
Response: Corrected in line 180
Less than 30% of male and female children experienced the same “what”?
Response: Corrected in line with 180-181
Line 178: what is that meaning of “almost even”
Response: Corrected in line 182
Line 172-199 is the same as table 2, this part needs to be refined.
Response: done in line 217-275
Line 228-235: the results found fathers and mothers needed the same education about exclusive breastfeeding, therefore this discussion part deviated from the research results.
Response: Done in line with 238-239
Discussion: it is better to divide into three paragraphs, 1st paragraph is the summary of the study, 2st and 3st paragraph discuss geography and parental education, respectively
Response: Done in line with 278-347
Line 224-225 could be moved into limitation.
Response: Done in line with 253
- Conclusions 7. Implication and Contribution are not sufficient, and should focus on the results of this current study.
Response: Done in line with 277-278
Line 247: double that, editing and language revisions of the whole manuscript are necessary.
Response Done
Reviewer 2 Report
In general the article gives interesting data about breastfeeding practices in Areas of Bangladesh. And after revision it might be accepted. It is worthwhile to be published in my opinion. However I have some remarks , because the paper is not always clear. For instance in the abstract icddr,b is used and not explained what it means. It is also not good to tell a reader that ISA did worse compared to GSA and then after that come with a conclusion contrary. The result is that ISA does better than GSA and after that you may gi it in introduction, aim of the study, methods , results and conclusion.
The introduction of the full article is ok,
Line 70 what is BDHS?? You must systematically first use a full name and put in brackets the abbreviations.
In my opinion Bangladesh is doing very good with 65% babies being exclusively breastfed under 6 months of age compared to f.i. The Netherlands 39 %, it was 18 % !
Line 119 what is PCA?
Question: has GSA a different ethical review board or is the ISA board good for both regions ?
Line 154: you have to explain 4+ANC
I was surprised that so many more fathers had no education compared to the adolescents mothers, can you explain that?
Line 206: what is a MNCH-FP program?
Line 192-196: to me it seems contradictory, the adolescent mothers from higher socio economic status had a lower risk of stopping earlier, I understand that, but then you say that adolescent mothers that used more ANC’s stop earlier, can you explain that since to me it is more logic that the higher socio economic class will also have more ANC visits.
Line 230: what is pre-lacteal feeding
Your conclusion doesn’t reflect the results of your study.
In general the English of the paper is not always clear to me. Maybe somebody can go over the whole paper to control the English.
Author Response
Reviewer 2:
In general the article gives interesting data about breastfeeding practices in Areas of Bangladesh. And after revision it might be accepted. It is worthwhile to be published in my opinion. However I have some remarks , because the paper is not always clear. For instance in the abstract icddr,b is used and not explained what it means.
Response: Done in line with16-17
It is also not good to tell a reader that ISA did worse compared to GSA and then after that come with a conclusion contrary. The result is that ISA does better than GSA and after that you may gi it in introduction, aim of the study, methods , results and conclusion.
Response This is done
The introduction of the full article is ok,
Line 70 what is BDHS?? You must systematically first use a full name and put in brackets the abbreviations.
Response: Done in line in 90
In my opinion Bangladesh is doing very good with 65% babies being exclusively breastfed under 6 months of age compared to f.i. The Netherlands 39 %, it was 18 % !
Response: We appreciate you concern
Line 119 what is PCA?
Response done in line 160
Question: has GSA a different ethical review board or is the ISA board good for both regions ?
Response: done in line 199
Line 154: you have to explain 4+ANC
I was surprised that so many more fathers had no education compared to the adolescents mothers, can you explain that?
Response: Done in line 302-307
Line 206: what is a MNCH-FP program?
Response: Done in line 286 with reference# 28.
Line 192-196: to me it seems contradictory, the adolescent mothers from higher socio economic status had a lower risk of stopping earlier, I understand that, but then you say that adolescent mothers that used more ANC’s stop earlier, can you explain that since to me it is more logic that the higher socio economic class will also have more ANC visits.
Response: Explain in line 313-321 (This results is not significant)
Line 230: what is pre-lacteal feeding
Response: done in line 337
Your conclusion doesn’t reflect the results of your study.
Response: Corrected in line 326-339
In general the English of the paper is not always clear to me. Maybe somebody can go over the whole paper to control the English.
Response: taken seriously and done throughout the paper.
Reviewer 3 Report
It is an interesting study and contributes to our understanding of factors affecting EBF practices among adolescent women, which will aid in improving the duration of EBF and subsequently female and nonadult care. Some revisions prior to publication must be made.
Title: seems to be a word missing, children follows women directly, not sure what you mean.
Abstract: it is problematic to have a lot of abbreviations, especially without explaining them all (i.e. icddr,b). A list of abbreviations at the end of the manuscript does not cover it. Make sure all readers are clear on what you mean. The grand WHY is missing in the abstract, but also in the paper. WHY do you want to know and WHY should the readers want to know the outcome of your research? How did it contribute?
Intro: line 40....reference to the information you used?
Materials and Methods: Figure 1 is of low quality and not legible. Please make sure the Figure is of sufficient DPI. The reference of this picture is missing. Make sure you add this.
Table 1: make sure you do not report just the p-value, but also the outcome of the tests.
Discussion: line 207, which covariates did you adjust for and why?
Line 211: what were the confounding variables and why did you need to adjust for them. What is the justification for this?
Line 212 and onwards, you compare your findings to single studies from other parts of the world. Who is Ghana or Morocco comparable to Bangladesh? Preferably, find more studies in this topic (there should be more) and make sure to describe how these studies are compatible with yours. If there are no more studies (which seems highly improbable), make sure you say this.
Line 224: your results are not insignificant, the differences may not be statistically significant. This is something different. Furthermore, it may not be due to sample size, but just because the difference is not statistically significant. This may be an interesting result which may further confound service and healthcare. Other factors that were not included in this study may have been contributing. Make sure to point this out.
Author Response
Reviewer 3:
It is an interesting study and contributes to our understanding of factors affecting EBF practices among adolescent women, which will aid in improving the duration of EBF and subsequently female and nonadult care. Some revisions prior to publication must be made.
Title: seems to be a word missing, children follows women directly, not sure what you mean.
Response: Changes in line 3.
Abstract: it is problematic to have a lot of abbreviations, especially without explaining them all (i.e. icddr,b). A list of abbreviations at the end of the manuscript does not cover it.
Response: Corrected in line 16
Make sure all readers are clear on what you mean. The grand WHY is missing in the abstract, but also in the paper. WHY do you want to know and WHY should the readers want to know the outcome of your research? How did it contribute?
Response: Corrected in line 94-109
Intro: line 40....reference to the information you used?
Response: References added in line 50-52 with references #9
Materials and Methods: Figure 1 is of low quality and not legible. Please make sure the Figure is of sufficient DPI. The reference of this picture is missing. Make sure you add this.
Response: Fig 1 is updated in line 142, this is also cited in references#25 in line 521-522
Table 1: make sure you do not report just the p-value, but also the outcome of the tests.
Response: The outcome of the test was indicated in Table 1 only in the cells of corresponding P-values with a “*” mark beside the p-value and the “*” mark implies that the corresponding covariate is significantly associated with EBF practice. This is also mentioned as a footnote of the table as “Note: * indicates that the results are significant at P-value < 0.05” in line 204 of the manuscript with track changes.
Discussion: line 207, which covariates did you adjust for and why?
Response: Adjusted variables line from 228-230. In this particular study, we have adjusted the cox-proportional hazard model with covariates child sex, mothers age at first birth, maternal education, paternal education, religion, place of delivery, mode of delivery, history of 4+ANC, and repeated pregnancy. Breastfeeding practices are not straight forward, it is highly vary with environment and buttress with complex network that consider personal, biological socio-cultural factors. Literature review helped us to identify these covariates associated with practice of EBF worldwide.
Line 211: what were the confounding variables and why did you need to adjust for them. What is the justification for this?
Response: The responses are same as previous covariates responses. In this particular study, we have adjusted the cox-proportional hazard model with confounding variables namely, child sex, mothers age at first birth, maternal education, paternal education, religion, place of delivery, mode of delivery, history of 4+ANC, and repeated pregnancy which is mentioned in the manuscript with track changes in lines 294-296 as “confounding variables (Child sex, Mother age at first birth, Maternal education, Paternal education, Religion, Asset Score, Place of delivery, Mode of delivery, History of 4+ ANC, Repeated pregnancy)”.
Line 212 and onwards, you compare your findings to single studies from other parts of the world. Who is Ghana or Morocco comparable to Bangladesh? Preferably, find more studies in this topic (there should be more) and make sure to describe how these studies are compatible with yours. If there are no more studies (which seems highly improbable), make sure you say this.
Response: references added in line with 311, 315
Line 224: your results are not insignificant, the differences may not be statistically significant. This is something different. Furthermore, it may not be due to sample size, but just because the difference is not statistically significant. This may be an interesting result which may further confound service and healthcare. Other factors that were not included in this study may have been contributing. Make sure to point this out.
Response: Done in line 325-333